# Dysregulation of the leukocyte signaling landscape during acute COVID-19

Isaiah R. Turnbull[1]*, Anja Fuchs[1], Kenneth E. Remy[2], Michael P. Kelly[3], Elfaridah P. Frazier[3], Sarbani Ghosh[1], Shin-Wen Chang[1], Monty B. Mazer[4], Annie Hess[1], Jennifer M. Leonard[1], Mark H. Hoofnagle[1], Marco Colonna[5], Richard S. Hotchkiss[1]

1 Departments of Surgery, Washington University School of Medicine, St. Louis, Missouri, United States of America, 2 Departments of Pediatrics, Washington University School of Medicine, St. Louis, Missouri, United States of America, 3 Departments of Orthopedic Surgery, Washington University School of Medicine, St. Louis, Missouri, United States of America, 4 Departments of Anesthesia, Washington University School of Medicine, St. Louis, Missouri, United States of America, 5 Departments of Pathology and Immunology, Washington University School of Medicine, St. Louis, Missouri, United States of America

* turnbulli@wudosis.wustl.edu

**Data Availability Statement:** The data underlying the results presented in the study are available from Flow Repository: https://flowrepository.org/id/FR-FCM-Z4PW.

## Abstract

The global COVID-19 pandemic has claimed the lives of more than 750,000 US citizens. Dysregulation of the immune system underlies the pathogenesis of COVID-19, with inflammation mediated tissue injury to the lung in the setting of suppressed systemic immune function. To define the molecular mechanisms of immune dysfunction in COVID-19 we utilized a systems immunology approach centered on the circulating leukocyte phosphoproteome measured by mass cytometry. We find that although COVID-19 is associated with wholesale activation of a broad set of signaling pathways across myeloid and lymphoid cell populations, STAT3 phosphorylation predominated in both monocytes and T cells. STAT3 phosphorylation was tightly correlated with circulating IL-6 levels and high levels of phospho-STAT3 was associated with decreased markers of myeloid cell maturation/activation and decreased ex-vivo T cell IFN-γ production, demonstrating that during COVID-19 dysregulated cellular activation is associated with suppression of immune effector cell function. Collectively, these data reconcile the systemic inflammatory response and functional immunosuppression induced by COVID-19 and suggest STAT3 signaling may be the central pathophysiologic mechanism driving immune dysfunction in COVID-19.

## Introduction

In December of 2019, COVID-19 syndrome emerged in the city of Wuhan, Hubei province of Central China. The novel corona virus SARS-CoV2 was rapidly recognized as the etiologic agent of COVID-19. Over the last 12 months, a combination of facile respiratory transmission, a high incidence asymptomatic carriers, and an immunologically naïve population have synergized to foment a global pandemic that has resulted in over 250 million cases of COVID-19 worldwide [1]. There have been more than 750,000 deaths in the US alone [2], comprising more than two out of a thousand US citizens, consistent with reports of an infection mortality rate ranging from 0.3–1%. SARS-CoV2 infectious disease spans the clinical spectrum from

**Funding:** This work supported by R35GM133756 (IRT) and R35GM126928 (RSH) provided by the National Institute of General Medical Sciences: https://www.nigms.nih.gov/ The funders had no role in study design, data collection and analysis, decision to publish, or preparation of the manuscript.

**Competing interests:** IRT, RSH, KER and MBM have pending intellectual property related to the ELISpot assay. This does not alter our adherence to PLOS ONE policies on sharing data and materials.

asymptomatic carriage to life-threatening critical illness. Approximately 5% of infected persons experience severe COVID-19 disease, clinically manifest as a viral sepsis syndrome defined by a dysregulated immune-inflammatory response and new onset organ failure [3–7]. The described pathophysiology of severe COVID-19 recapitulates the last 30 years of sepsis research: a fulminant cytokine dyscrasia overlying functional immunosuppression, coagulopathy and thrombosis-mediated organ damage, neurologic complications including delirium and ICU psychosis, and a risk of long-term, persistent illness [8].

Early reports characterized COVID-19 as a cytokine-storm syndrome. This hypothesis was based on data demonstrating elevated levels of canonical inflammatory mediators such as IL-6 and CXCL-10, suggesting that a cytokine-induced systemic inflammatory response was driving the pathophysiology of the disease [9–12]. Subsequent studies elucidated a more complex immunophenotype for COVID-19, with profound changes in both myeloid and lymphoid leukocyte populations, including mobilization of immature myeloid cells from the bone-marrow [13–15], an increase in circulating B-cell plasmablasts and an overall lymphopenia including CD4, CD8, and invariant (γδ) T cells. These changes in leukocyte populations were associated with an increase in immature "low-density" neutrophils, an increase in immunosuppressive HLA-DR$^{lo}$ monocytes and a shift toward a broadly "activated" T cell phenotype, which based on correlation analysis were driven by the circulating inflammatory cytokinemia [16–19]. Functional studies directly assessing leukocyte function demonstrated that like severe sepsis, severe COVID-19 is associated with an overall suppressed immunotype [20, 21]. Our group reported that severe COVID-19 was associated with defects in T cell production of IFN-γ and monocyte TNF-alpha production [22] and others reported weak CD8 T cell responses to COVID-19 antigen and impaired DC function [23]. Taken together, the data present a complex picture of immune dysfunction during COVID-19 with high levels of circulating inflammatory mediators, neutrophilia, and partial T cell activation, but functional immunosuppression.

To better characterize the effect of COVID-19 on the immune system we sought to define the intracellular signaling milieu of circulating leukocytes during acute COVID-19. We deployed a mass cytometry assay to measure the leukocyte phosphoproteome in moderate and severely ill COVID-19 patients and correlated changes in the intracellular signaling profile with circulating cytokine levels and measures of cellular activation and function. We found that COVID-19 is associated with a significant dysregulation of the leukocyte signaling landscape, with broad based increase in signaling phosphoprotein levels in all circulating leukocyte populations. In myeloid cell populations, surface markers of activation and maturation were negatively correlated with activation of ERK, STAT3, STAT1 and CREB. Similarly, in CD4 and CD8 T cells, high levels of STAT3 phosphorylation was correlated with defects in ex-vivo IFN-γ production. Taken together these data begin to reconcile the COVID-19 induced inflammatory cytokinemia with the well described functional immunosuppression documented in critically ill COVID-19 patients [9, 14, 17, 18, 21–27].

## Methods

### Study design and recruitment

Patients were recruited from April 2020-November 2020 at Missouri Baptist Medical Center and Barnes-Jewish Hospital. Symptomatic patients over age 18 with a pending, clinically indicated SARS-CoV2 nasopharyngeal PCR test were approached for consent. All subjects or their legally authorized representatives provided written informed consent for this study. Subjects with a subsequent negative COVID-19 test were excluded. Blood samples were obtained from peripheral venipuncture or from existing venous catheters. All blood samples included in this analysis were obtained within 72 hours of hospital presentation. 63 subjects are included. 43

subjects had severe COVID-19 disease requiring admission to the intensive care unit (ICU). 20 subjects had moderate disease requiring inpatient hospitalization but no ICU care. Demographic and clinical data was abstracted from the electronic medical record. Concurrently, a cohort of healthy donor subjects were recruited. Healthy donors had no self-reported acute illness or history of chronic infection, autoimmune disease, malignancy or organ transplantation. For all samples, blood was collected into heparinized vacutainers stored at room temperature. All samples were processed within 4 hours of phlebotomy. Study protocols were approved by the Washington University in St. Louis Institutional Review Board (approval # 202003085) or the Missouri Baptist Medical Center Institutional Review Board (approval # 1132).

## Sample processing, staining and mass cytometry acquisition

1 mL of whole blood was mixed with 1.4 volumes of Proteomic Stabilizer (Smart Tube Inc, San Carlos, CA), incubated at room temperature for 10 minutes, then stored at -80˚ Celsius until analysis. At the time of analysis samples were thawed and red blood cells were lysed using Thaw-Lyse buffer (Smart Tube Inc, San Carlos, CA) and permeabilized for staining with Max-Par Barcoding Perm Buffer (Fluidigm Inc, San Francisco, CA) following manufacturers' protocols. Cell numbers were enumerated using a hemocytometer and 3x10^6 cells were transferred for staining. The enumerated total cells/ml were used to calculate the cell frequencies. Individual samples were first barcoded by incubation with Cell-ID 20-Plex Pd Barcoding Kit (Fluidigm) following manufacturers protocol. Up to 10 samples were pooled, washed x2 with CyFACS buffer (Thermo Fisher, Waltham, MA) then resuspended in 0.5 mL CyFACS. Fc Receptors were blocked by adding 45 uL TruStain FcX (Biolegend) at RT for 10 min. Surface antibody cocktail was then added (see S1 Table "Antibody panels for surface and intracellular staining") and sample incubated for 1 hour on ice. Cells were then washed in 10 mL of CyFACS buffer x1.

For intracellular staining, the cell pellet was resuspended in -20˚C methanol to a final concentration of 5x10^6 cells/ml, incubated at -20˚C for 30 min, then washed 2x in 10 mL of ice-cold CyFACS buffer. Cells were resuspended in 1mL CyFACS buffer then incubated for 60 minutes on ice with intracellular antibody cocktail. Cells were washed x2 with 10 mL CyFACS buffer, then resuspended in FACS buffer containing 2% PFA and Cell ID Intercalator Solution (Fluidigm, San Francisco, CA) following manufacturers protocol for DNA staining.

Cells were analyzed on a Fluidigm CyTOF 2 Mass Cytometer. Samples were first resuspended in water+10% EQ Four Element Calibration Beads (Fluidigm) then acquired with an event rate of ~500 events/sec. CyTOF data were analyzed using the FlowJo software platform (BD, Franklin Lakes, NJ).

All antibodies were commercially conjugated to heavy metal isotopes except for clones B1.1, WM53, ICRF44, VI-PL2 and 7C9. For these clones unconjugated antibodies were purchased and conjugated using MAXPAR Antibody Labeling Kit (Fluidigm, San Francisco, CA) following manufacturers protocols.

**Soluble molecule determination.** For plasma studies, cellular elements of whole blood were pelleted by centrifugation at 1000 xg for 7 minutes at room temperature; the soluble phase was aspirated, aliquoted and stored at -80˚C. Cytokine and chemokine levels were determined by multiplex bead array (Cytokine Human Magnetic 35 Plex Panel, ThermoFisher Scientific) following manufacturers protocol. Samples were acquired on a Luminex FLEXMAP 3D instrument system.

## Enzyme-Linked Immunospot (ELISpot) assay

Peripheral blood mononuclear cells (PBMCs) were isolated from 5 mL fresh whole blood as previously described [22]. Total number of PBMCs was determined using a Vi-Cell™ viability

analyzer (Beckman Coulter, Brea, CA, USA). Flow cytometry was performed for cell typing, staining for CD3, CD4, CD8 and CD14. Detection of *ex vivo* production of IFN-γ was assessed by ELISpot using precoated plates (ImmunoSpot by Cellular technology Limited (CTL), Cleveland, OH, USA) as per manufacturer's protocols. Cells were incubated in serum-free media (CTL) with 500 ng/mL of anti-CD3 (Bio-legend clone HIT3a) with 2.5 μg/mL of anti-CD28 (Bio-legend clone CD28.2) antibodies in a total well volume of 200 μg for 2.5 x 10⁴ PBMCs. Following overnight incubation at 37˚C in 5% $CO_2$, biotinylated detection antibody, streptavidin bound alkaline phosphatase and developer solution were applied as per manufacturer instructions. ELISpot analysis was performed using a CTL series 6 ImmunoSpot Universal Analyzer with Immunospot 7.0 professional software (CTL Analyzers, Shaker Heights, OH).

## Data analysis

Cell number, cell frequency, subpopulation frequency, signal intensities and soluble molecule levels were compared across healthy, moderate COVID-19 and severe COVID-19 cohorts by non-parametric Kruskall-Wallis ANOVA, followed by correction for a false discovery rate (FDR) using the method of Benjamini, Krieger and Yekutieli. Features with <1% FDR were considered significant. Post-hoc bivariate comparison was done by Dunn's multiple comparison test. To correct for batch effects across CyTOF runs, signal intensities were normalized to a common reference sample. Blood was isolated from a healthy individual at a single time point, aliquoted and frozen. The reference sample was a healthy donor blood sample was included in every batch as an internal control across CyTOF run. This sample was included in every as a reference standard. Signal intensity for experimental samples were normalized by dividing the experimental sample geometric mean by the reference sample geometric mean for each feature. Normalized signal intensities were used for downstream analysis. ELISpot data was analyzed by Kruskall-Wallis ANOVA. Post-hoc bivariate comparison was done by Dunn's multiple comparison test.

Correlation between CyTOF features and soluble markers and ELISpot results was done with Spearman Rank Order Correlation.

Kruskall-Wallis and Dunn's test were run on SPSS V12 (IBM Corporation, Armonk, NY). Benjamini, Krieger and Yekutieli Analysis was done in Prism (Graphpad Software, San Diego, CA).

## Results

### Mass-cytometry defines the immunologic pathways induced by COVID-19

To define the effects of SARS-COV2 infection on the intracellular signaling environment of circulating leukocytes, we recruited a cohort of 63 subjects with COVID-19. COVID-19 severity was defined using modified WHO disease severity criteria. Critically ill patients requiring intensive care were characterized as "Severe"; patients requiring inpatient care but not intensive care were defined as "Moderate". Blood was isolated from 20 subjects with moderate illness and 43 subjects with severe illness. All samples were obtained within 72 hours of hospital presentation. A cohort of healthy donors served as controls. Peripheral blood was analyzed by mass cytometry. For subsets of these patients, plasma cytokine levels and T cell interferon γ production by ELISpot assay were assayed (Fig 1A). ELISpot and plasma cytokine levels were measured when laboratory staff was available to perform these assays and when there was adequate sample volume. Demographic characteristics of all cohorts are shown in S2 Table "Cohort Demographics and Description". Subjects with severe COVID-19 had a longer length of stay as compared to subjects with moderate COVID-19. 58% of patients with severe COVID-19 required mechanical ventilation, and within the severe COVID-19 cohort,

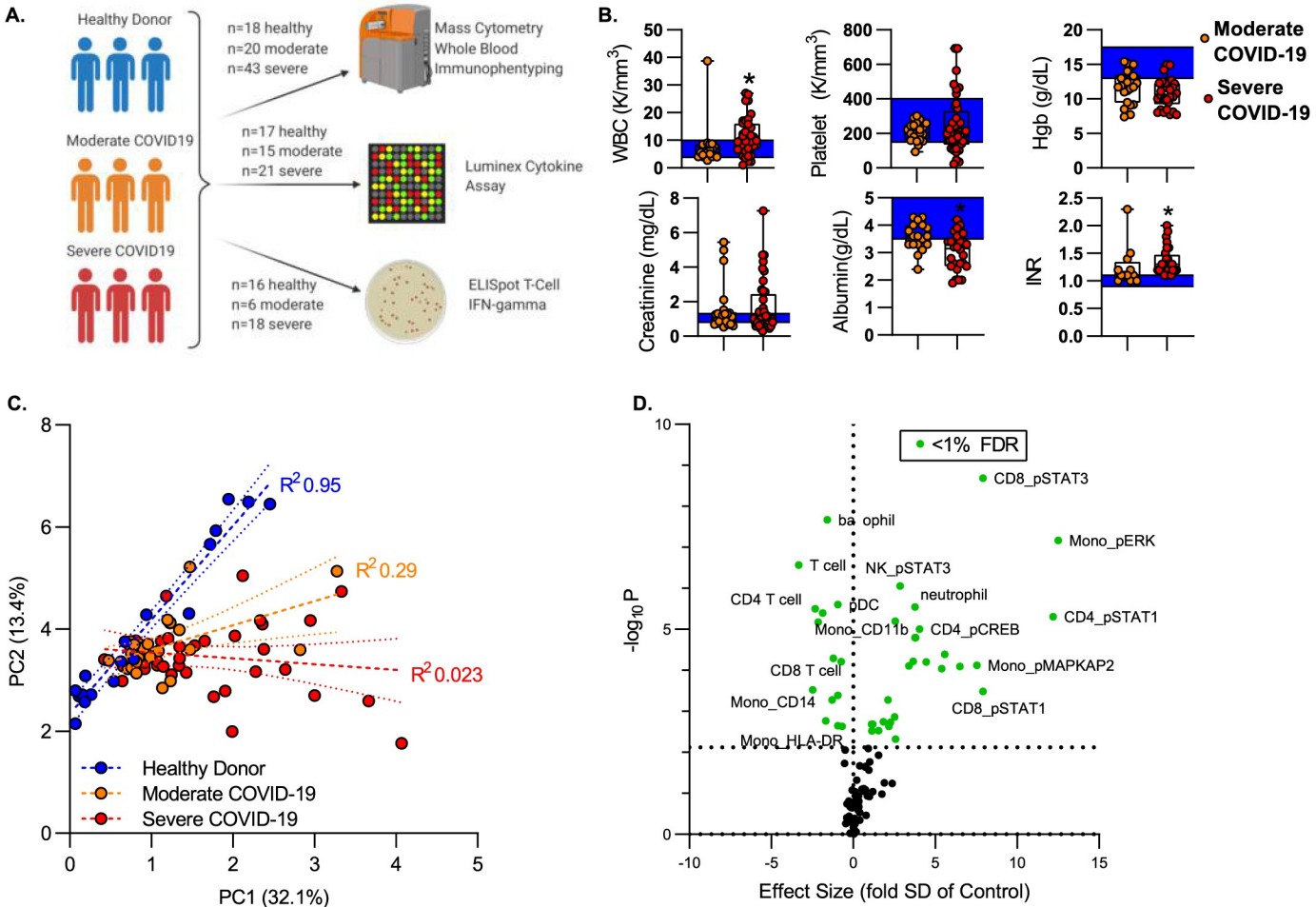

**Fig 1. Mass cytometry to define the molecular mechanisms of COVID-19. A)** Study overview. Peripheral blood was isolated from healthy donors or patients with COVID-19. All subjects were recruited within 72 hours of hospital admission. "Moderate" was defined as acute illness requiring hospitalization; "Severe" as subjects with critical illness requiring ICU care. Whole blood was fixed and stored for batch CyTOF analysis. Plasma was stored for soluble marker analysis. Peripheral blood mononuclear cells were isolated by density gradient centrifugation for T cell function assay by ELISpot. **B)** Clinical laboratory values of patients with COVID-19. Box plots represent 25-75$^{th}$ percentile with line defining group median, with all data-points overlying. Error bars extend from minimum to maximum values. Shaded area represents the normal value reference range. $^*$p<0.05, $^{**}$p<0.01, $^{***}$p<0.001, $^{****}$p<0.0001 vs moderate COVID-19 by Mann-Whitney U test. **C)** Principal Component Analysis of 95 cellular featured define by CyTOF. Dashed lines represent linear regression line of the first two principal components for each cohort (healthy, moderate COVID, severe COVID); dotted lines show the 95% confidence intervals. **D)** Volcano plot of features defined by CyTOF. Effect size defined as difference of means between severe COVID vs. healthy, divided by standard deviation of healthy. P-value was measured by nonparametric (Kruskal-Wallis) ANOVA across cohorts. Significance defined as false discovery rate (FDR)< 1% by Benjamini, Krieger and Yekutieli FDR<1% = p<0.0058 (dotted line). Features that vary significantly across groups shown in green.

mortality was 37%. Consistent with prior studies, clinical laboratory values demonstrated that COVID-19 was associated with leukocytosis (Fig 1B). We also identified anemia in both moderate and severe COVID-19 patients. Severe COVID-19 was associated with elevated serum creatinine, although there was no significant difference between subjects with moderate vs. severe disease. Both moderate and severe COVID-19 was associated with a coagulopathy manifest as an elevated INR; further, subjects with severe COVID-19 had increased INR as compared to moderately ill subjects (Fig 1B). Complete clinical laboratory profiles are shown in S3 Table "Clinical Lab Data".

We used mass-cytometry to define the immunologic features associated with COVID-19 disease. We analyzed blood using a panel of heavy-metal conjugated antibodies recognizing

both cell surface antigens and intracellular signaling molecules coupled with mass cytometry (See S1 Table). Cell type was defined based on cell surface phenotype by on manual gating using canonical markers (S1 Fig "Gating Strategy for Mass Cytometry"). Within each population, we measured the signal intensity of 15 intracellular signaling molecules which are key mediators of T cell and monocyte response to invading pathogens. Altogether, we defined 95 cellular features that identify the dominant response of immune effector cells to COVID-19 infection (S4 Table "Immunologic Features Defined by CyTOF"). To understand the overall structure of the CyTOF data we performed factor reduction by Principal Component Analysis. Fig 1C shows an X-Y plot of the first two principal components, which together account for 45% of the overall variance in the dataset. Healthy subjects were tightly arrayed along a common line. In contrast, subjects with COVID-19 formed distinct clusters with severely ill subjects having a higher variance than that of moderately ill subjects. These PCA plots demonstrate that our CyTOF features can segregate healthy donors from moderately and severely ill COVID-19 patients. To identify specific features associated with COVID-19 disease, we used the non-parametric rank-order Kruskall-Wallis test to compare the distribution of each feature across groups (healthy controls, moderate COVID-19, severe COVID-19) and corrected for a 1% false discovery rate (FDR) using the Benjamini, Krieger and Yekutieli approach. After false-discovery correction, we identified 43 features with differential variance across condition (healthy, moderate COVID-19 and severe COVID-19, see S4 Table). To visualize the interaction between p-value and effect size we first calculated an effect size for each feature. Effect size was estimated using a modified z-score, calculated as the difference in means between subjects with severe COVID-19 and healthy controls, divided by the standard deviation of the healthy control population. Fig 1D shows volcano plot of effect size vs. p-value, with features reaching an FDR of 1% shown in green.

To define the immunologic effects of COVID-19, on the absolute numbers of different immune effectors comprising innate and adaptive immunity, we first evaluated circulating leukocyte frequency and number. Using canonical surface markers to define leukocyte types, we measured the frequency and absolute number of circulating leukocyte cell populations. Consistent with prior studies, we found that COVID-19 was associated with an increase in neutrophil frequency, although there was a trend toward increased neutrophil numbers, this trend did not reach a statistically significant threshold. We did measure a significant decrease in frequency of both innate and adaptive leukocytes including monocytes, CD4 T cells, CD8 T cells, plasmacytoid dendritic cells (pDC) and NK cells. These differences in frequency translated to absolute leukopenia during COVID-19 for most of the cell populations measured, including monocytes, CD4 and CD8 T cells and pDC (Fig 2). We detected no differences in B-cell frequency or number.

## Dysregulated signaling landscape induced by acute COVID-19

To define the effects of COVID-19 on the intracellular signaling landscape of circulating leukocytes we measured a panel of signaling phosphoproteins in neutrophils, classical monocytes, NK cells, CD4 and CD8 T cells. For each sample, leukocyte populations were manually gated based canonical markers (S1 Fig) and we measured signal intensity of phosphoproteins within each population (S4 Table). To visualize the overall structure of the data we prepared a heatmap of phosphoprotein signaling intensities in each population for all samples, then clustered based on sample and row using hierarchical clustering (Fig 3). In this histogram row color is normalized by row min-max which exaggerates small differences. Across samples, healthy donor clustered tightly and were distinct from subjects with COVID-19, although moderate and severe COVID-19 were intermixed (Fig 3). Visual review of the heatmap did identify

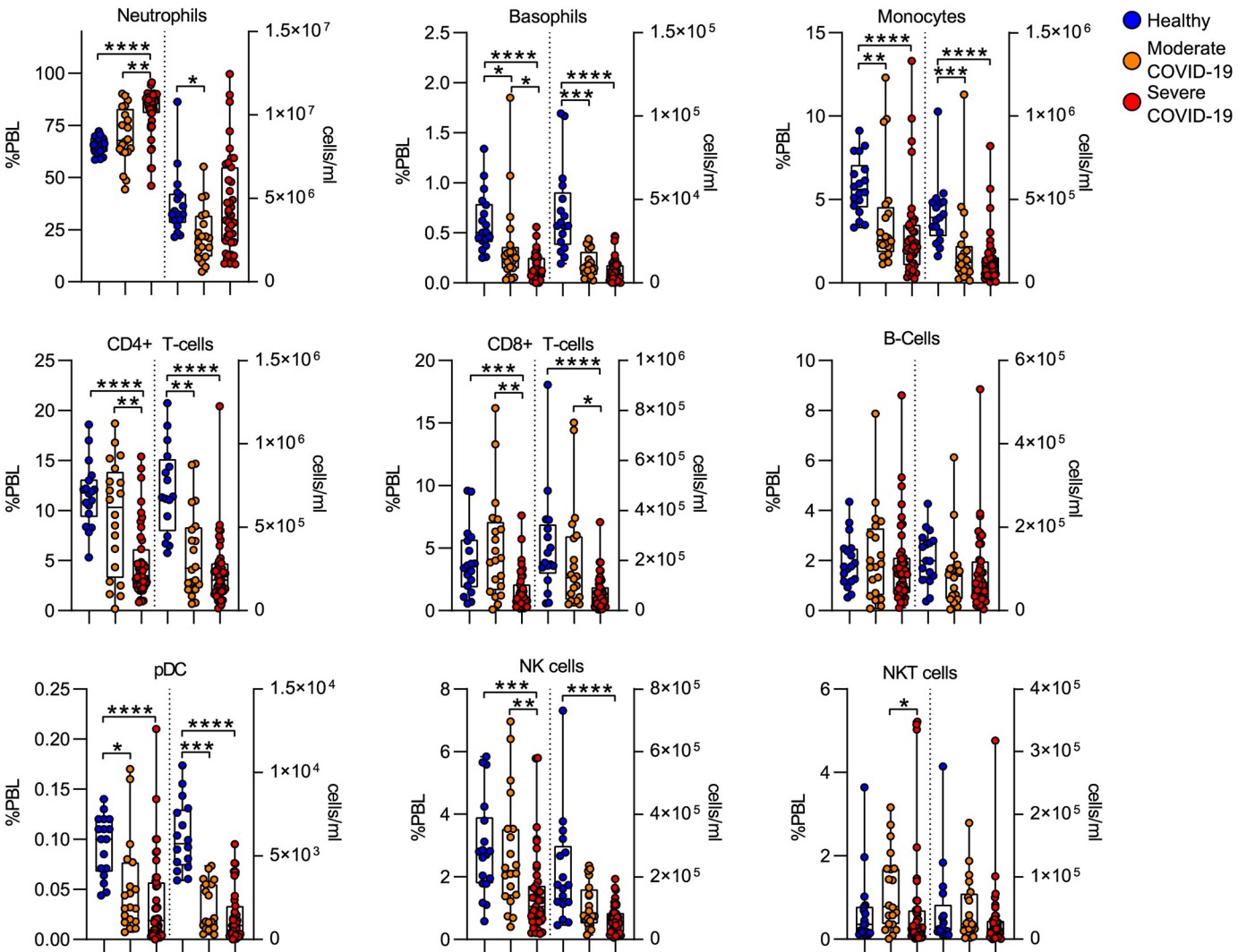

**Fig 2. Effect of COVID-19 on circulating leukocyte populations.** Leukocyte populations were manually gated based on canonical surface marker expression. The frequency of CD45+ peripheral blood leukocytes (%PBL) and the absolute concentration of cells (cells/ml) were calculated. Box plots represent 25-75th percentile with line defining group median, with all data-points overlying. Error bars extend from minimum to maximum values. *p<0.05, **p<0.01, ***p<0.001, ****p<0.0001 by Dunn's multiple comparisons test after nonparametric (Kruskal-Wallis) ANOVA with correction FDR<1% (Benjamini, Krieger and Yekutieli).

cluster of features that did vary across healthy donors, but subsequent statistical analysis did not measure any significant variation in these features across the groups. These clusters are features with low signal intensity which is visually exaggerated by the color mapping but does not represent significant variation. Hierarchical clustering across features demonstrated shared covariance of phospho- STAT1 and STAT3 (pSTAT1, pSTAT3) signals. pSTAT1 and pSTAT3 levels clustered together across cells types including neutrophils, monocytes, CD4 T cells and CD8 T cells, suggesting that STAT1 and STAT3 phosphorylation levels may distinguish the normal cellular signaling pathways operative in healthy donors versus COVID-19 induced altered signaling pathways and subjects with COVID-19 (Fig 3).

To further characterize the effect of COVID-19 on the leukocyte phosphoprotein landscape we measured the magnitude and direction of the signaling effect induced by COVID-19. Effect size was defined by a modified z-score calculated as the difference in median signaling intensity

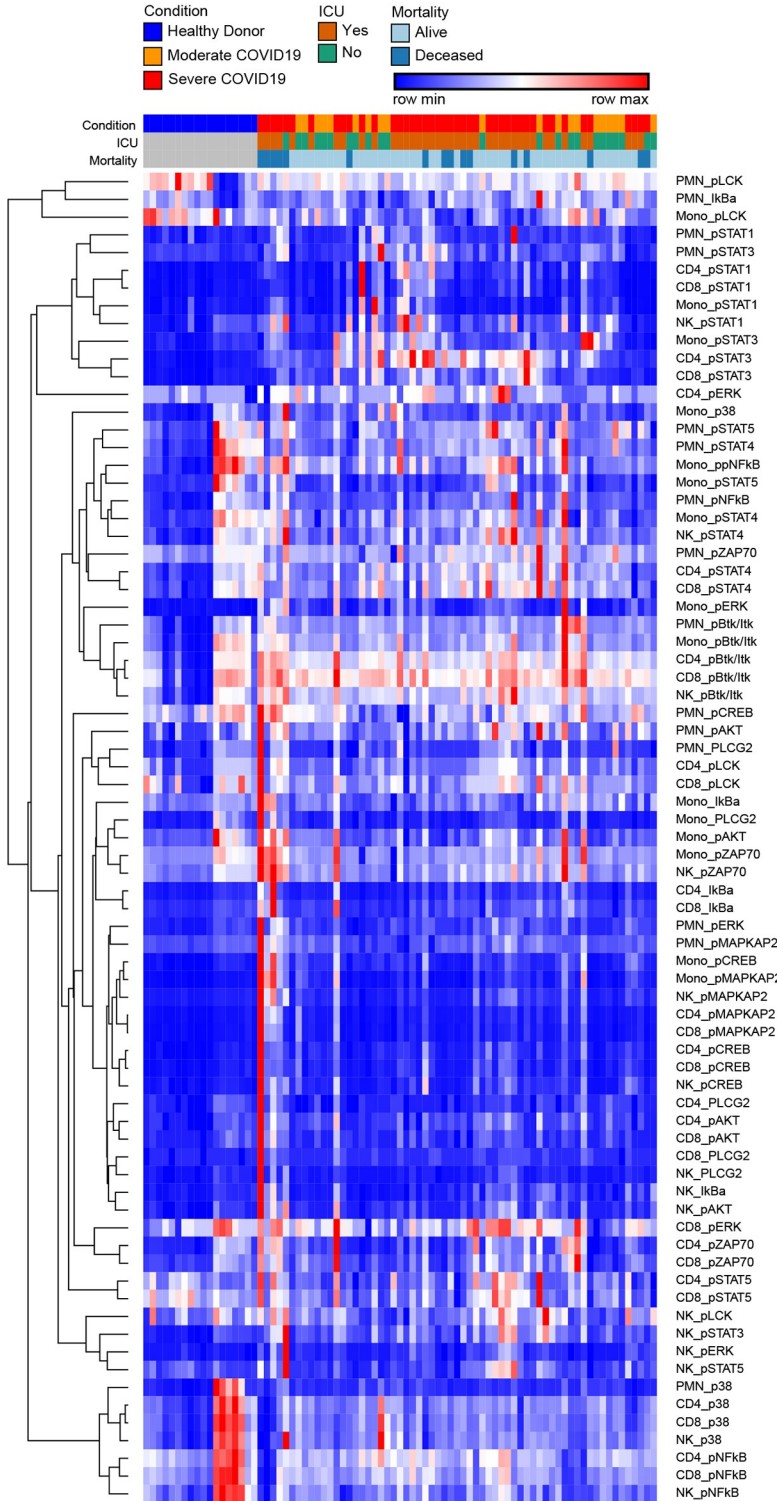

**Fig 3. Heatmap of hierarchically clustered signaling phosphoprotein levels.** Columns are individual samples and rows are cellular phosphoprotein features as labelled. Values are hierarchically clustered on columns and rows. Color values are normalized to row min-max.

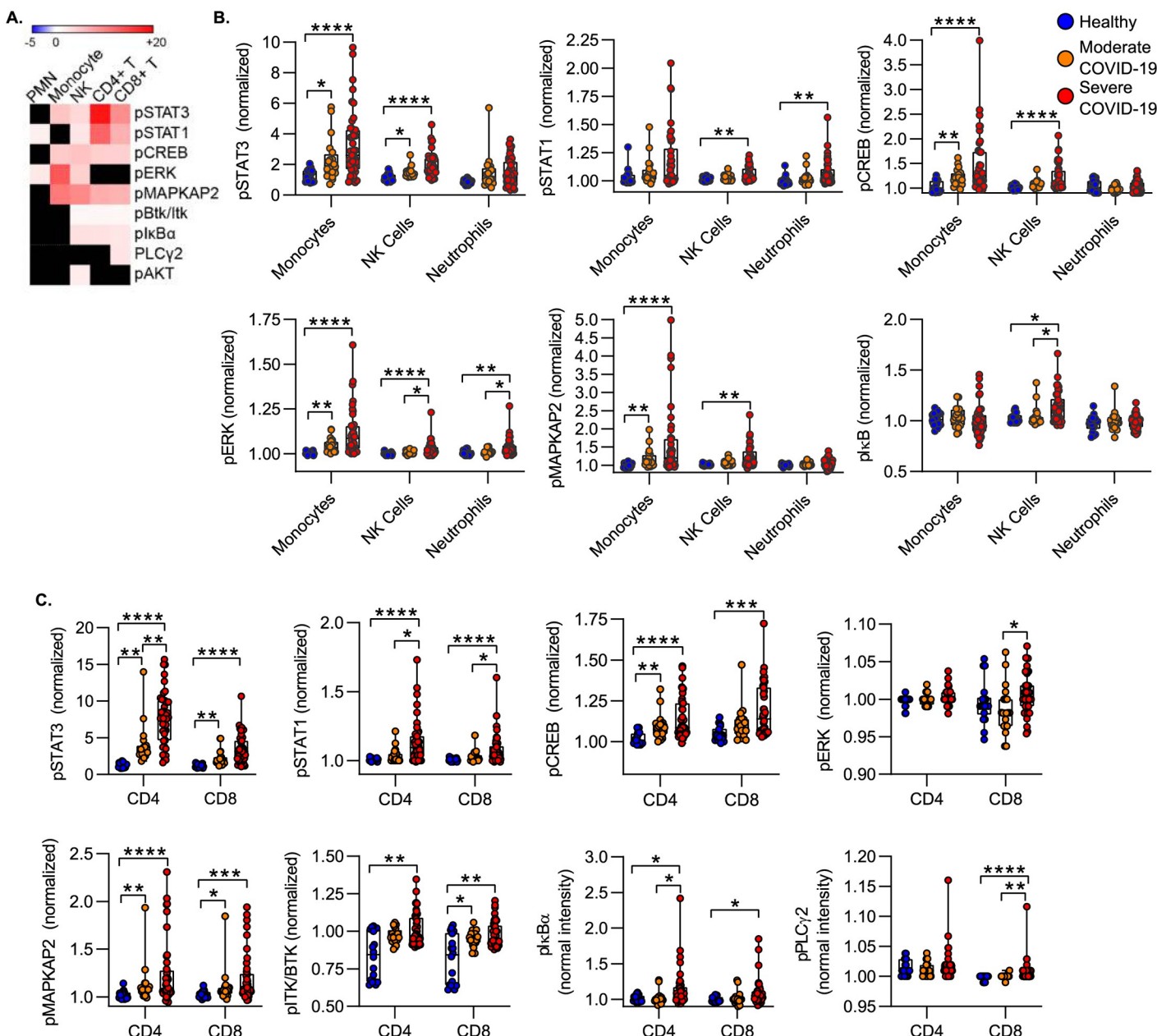

**Fig 4. Effect of COVID-19 on circulating leukocyte phosphoproteome. A)** Heat map of effect size of severe COVID-19 vs. healthy controls. Effect size defined as difference of means of severe COVID vs. healthy, divided by standard deviation of healthy. Black squares are features which are not significantly different. **B-C)** Phosphoprotein levels in innate **(B)** and adaptive **(C)** leukocytes. To correct for CyTOF batch effects, all signal intensities are normalized to a common reference control and shown as fold-change of the reference control. Box plots represent 25-75th percentile with line defining group median, with all data-points overlying. Error bars extend from minimum to maximum values.*p<0.05, **p<0.01, ***p<0.001, ****p<0.0001 by Dunn's multiple comparisons test after nonparametric (Kruskal-Wallis) ANOVA with correction FDR<1% (Benjamini, Krieger and Yekutieli).

between severe COVID-19 and healthy controls, divided by the standard deviation in the control population. Fig 4 is a heatmap of the signaling effect size induced by severe COVID vs. healthy control (black squares are features which did not reach a statistically significant threshold). Overall, COVID-19 was associated with wholesale increases in signaling phosphoprotein levels. Consistent with hierarchical clustering data, we measured the largest effect sizes measured

for STAT3 and STAT1 (Fig 4A). As compared to healthy donors, we found only increased levels of signaling molecule phosphorylation, and in no case was COVID-19 associated with decreased phosphoprotein signal intensity as compared to healthy donors (Fig 4A).

Within the innate immune compartment, we evaluated changes in signaling phosphoprotein levels in monocytes, NK cells and neutrophils. In monocytes, we found that as compared to healthy donors both moderate and severe COVID-19 was associated with increased phosphorylation of STAT3, CREB, MAPKAP2 and ERK; there was no difference between moderate and severe COVID-19. We saw broader but less robust activation in NK cells, including elevated levels of phosphorylated STAT1, STAT3, ERK, MAPKAP2, CREB and IκB. In neutrophils, phospho-STAT1 and phospho-ERK distinguished healthy donors for patients with COVID-19 (Fig 4B).

Among T cells, we detected broad increases in signaling protein phosphorylation in both CD4 and CD8 T cells. Notably in both populations, STAT3 phosphorylation predominated, with a >7-fold increase in phospho-STAT3 in CD4 T cells and a >3-fold increase in CD8 T cells in severe COVID-19 subjects vs. healthy donors (Fig 4C). Phospho-STAT3 levels in CD4 and CD8 T cells distinguished between healthy donors, moderate COVID-19, and severe COVID-19. STAT1, MAPKAP2, CREB and IκBα were also increased in severe COVID-19 in both T cell populations. In both CD4 and CD8 T cells, MAPKAP2 and ITK/BTK phosphoprotein levels were increased in moderate vs. healthy donors. In contrast with the innate immune cell populations, we did not detect significant levels of ERK phosphorylation in CD4 or CD8 T cells (Fig 4C).

## During COVID-19 inflammatory cytokines are associated with STAT3 and ERK phosphorylation in circulating leukocytes

To define the interactions between the leukocyte phosphoproteome and the systemic inflammatory response during COVID-19, we measured plasma levels of 36 cytokines and chemokines in a subset of the subjects (S5 Table "Cytokines Levels"). Hierarchical clustering identified 4 cytokine clusters, with inflammatory mediators including IL-6, IL-8, CXCL-10 and HGF clustered together (S2 Fig "Heatmap of cytokine data"). Consistent with prior studies, we found that IL-6, IL-8, IL-1RA, CXCL-10 distinguished healthy controls from both moderate and severe COVID-19 cases (Fig 5A). In addition, HGF and MCP-1 distinguished moderate from severe COVID-19 (Fig 5A).

To understand the pathways driving cellular activation in neutrophils, monocytes and NK cells, we evaluated the relationship between phosphoprotein levels and circulating cytokine levels by measuring the correlation between phosphoprotein levels and cytokine levels across all samples (healthy, moderate COVID-19 and Severe COVID-19). Fig 5B shows a heatmap of Spearman correlation coefficients and 5C shows scatter plots of a subset of highly correlated features. In monocytes and NK cells, phospho-STAT3 levels were strongly correlated with circulating levels of IL-6 and CXCL10. In neutrophils phospho-STAT1 was highly correlated with IL-6 and CXCL10 as were levels of phospho-STAT1 in neutrophils. In contrast, monocyte phospho-ERK levels were most correlated with plasma levels of IL-8 (Fig 5C).

We applied a similar approach to define the signaling pathways activated in circulating CD4 and CD8 T cells. We again sought to correlate protein phosphorylation with circulating cytokine levels. In both CD4 and CD8 T cells, STAT1 and STAT3 phosphorylation were strongly correlated with IL-6, CXCL10, IL-8 and IL-1RA levels (Fig 5D/5E). Consistent with the canonical signaling downstream of the IL-6 receptor [28–30], CD4 T cell pSTAT3 was most highly correlated with circulating IL-6 levels, with a Spearman R of 0.85. This suggest that circulating IL-6 drives CD4 T cell STAT3 activation during acute COVID-19.

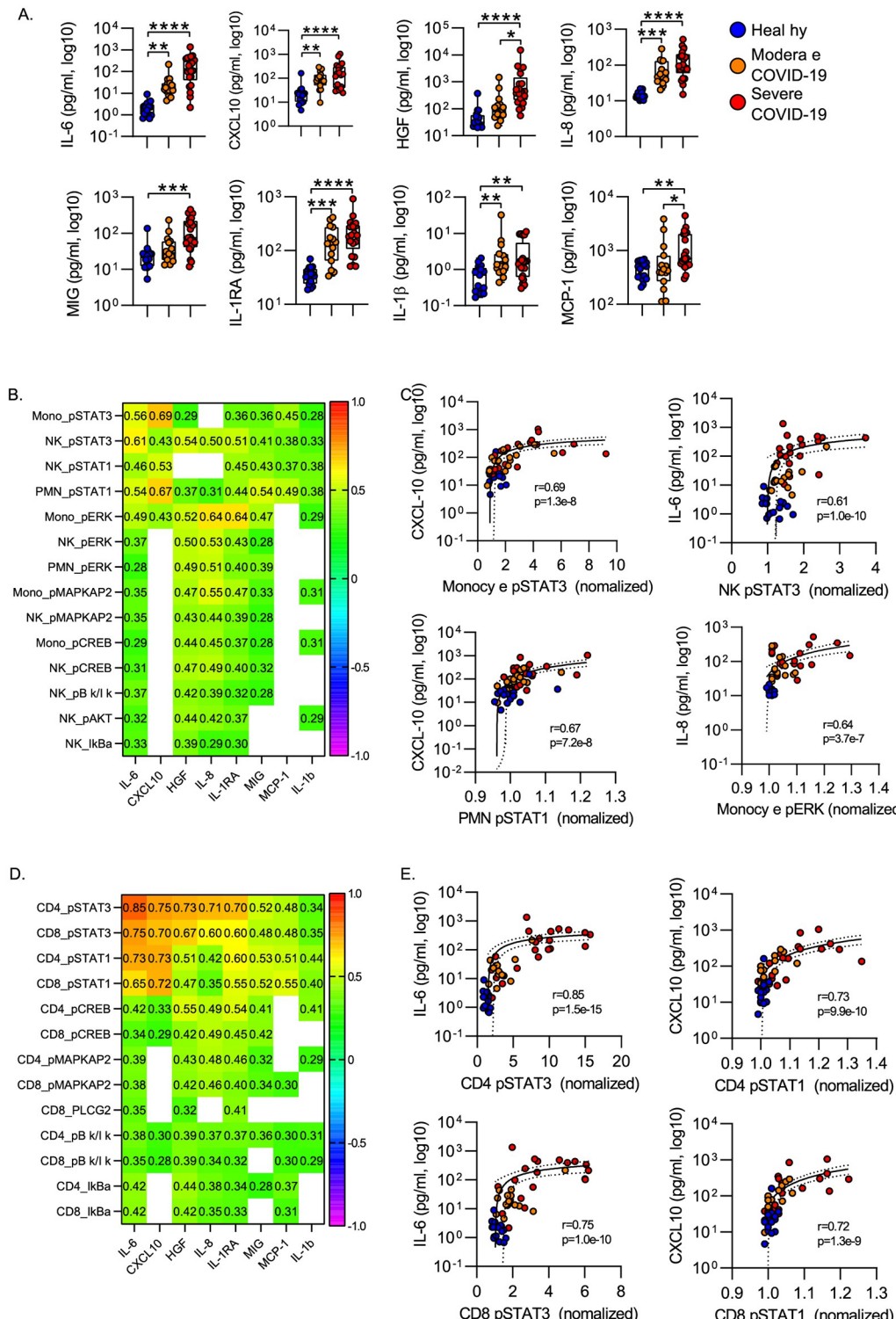

**Fig 5. Correlation of Inflammatory cytokines with leukocyte phosphoproteome. A)** Plasma cytokines were measured in a multiplex cytokine assay. *p<0.05, **p<0.01, ***p<0.001, ****p<0.0001 by Dunn's multiple comparisons test after nonparametric (Kruskal-Wallis) ANOVA with correction for FDR<1% (Benjamini, Krieger and Yekutieli). Box plots represent 25-75th percentile with line defining group median, with all data-points overlying. Error bars extend from minimum to maximum values. **B)** Heatmap of Spearman correlation between phosphoprotein signal intensity and plasma cytokine levels for innate immune cells. **C)** Correlation between cellular phosphoprotein levels and plasma

cytokine for highly correlated featured in innate immune cells. **D)** Heatmap of Spearman correlation between phosphoprotein signal intensity and plasma cytokine levels for adaptive immune cells. **E)** Correlation between cellular phosphoprotein levels and plasma cytokine for highly correlated featured in adaptive immune cells. **B/D**: Cell values show Spearman R for correlation with p<0.05. **C/E**: Plots display correlation results from Spearman correlation test and semilog regression curve (solid line) with 95% confidence intervals (dotted lines).

## Signaling phosphoprotein levels correlate with immune cell functional metrics

Severe COVID-19 is associated with a dysregulation of the myeloid cell compartment characterized by increased numbers of immature neutrophils in circulation, a shift to HLA-DR-lo monocytes and defects in the function of mature neutrophils in the circulation. Prior work has characterized significant heterogeneity in the circulating neutrophil compartment induced by COVID-19, including accumulation of neutrophils expressing decreased levels of HLA-DR, CD15, CD11b. Consistent with this, we detected decreased mean surface expression of HLA-DR and CD15 on the neutrophils during both moderate and severe COVID-19 as compared to healthy donors. CD11b significantly decreased in severe COVID-19 as compared to both healthy donors and subjects with moderate COVID-19 disease. We also evaluated the relationship between changes in phosphoprotein levels and surface marker expression. In neutrophils, we found that surface CD11b levels were negatively correlated with intracellular phospho-ERK (pERK) levels, with lower mean levels of CD11b in samples with higher pERK levels. Surface expression of CD15 on neutrophils was more tightly (and negatively) correlated with phospho-STAT1 levels (Fig 6B). Similarly, we found lower levels of CD11b, HLA-DR, CD14 and CD33 on the monocyte surface (Fig 6C). In monocytes, we again measured a negative correlation between increased signaling phosphoprotein levels and surface marker expression. In monocytes, CD11b levels were negatively correlated with phospho-STAT3 levels and HLA-DR levels correlated with phospho-CREB levels (Fig 6D).

We have previously shown that critical illness including COVID-19 is associated with adaptive immune dysfunction as measured by defects in T cell IFN-γ production in response to *ex vivo* T cell receptor stimulation [22]. Consistent with these data, we found that COVID-19 was associated with attenuated production of IFN-γ after stimulation of peripheral blood mononuclear cells with agonistic antibodies against CD3/CD28, suggesting a defect in T cell function during COVID-19 as compared to healthy controls (Fig 6E). We then assessed the relationship between phosphoprotein levels and T cell function. We found that increased pSTAT3 levels in both CD4 and CD8 T cells was associated with decreased production of IFN-γ in response to CD3/CD28 stimulation, suggesting increased STAT3 activation as a mechanism for defects in T cell function during COVID-19. As both CREB and PLC-γ2 can be involved with T cell activation, we also evaluated the relationship between the levels of phospho-CREB and phospho-PLC-γ2 and IFN-gamma production. There was no significant correlation between CD4 or CD8 pCREB and IFN-γ production. We did find a statistically significant correlation between CD4 phospho-PLC-γ2 levels and IFN-gamma production (Spearmans R = -0.46, p<0.03), suggesting a negative role for baseline activation of the PLC-g2 pathway in the response of the cells to subsequent CD3/CD28 ligation.

## Discussion

We deployed a systems biology approach centered on a mass-cytometry phosphoproteome assay combined with multiplexed measurement of circulating cytokine levels and ex-vivo stimulated T cell function by ELISpot to define the effect of COVID-19 on the intracellular signaling landscape of circulating leukocytes during acute COVID-19. We find that COVID-19 is

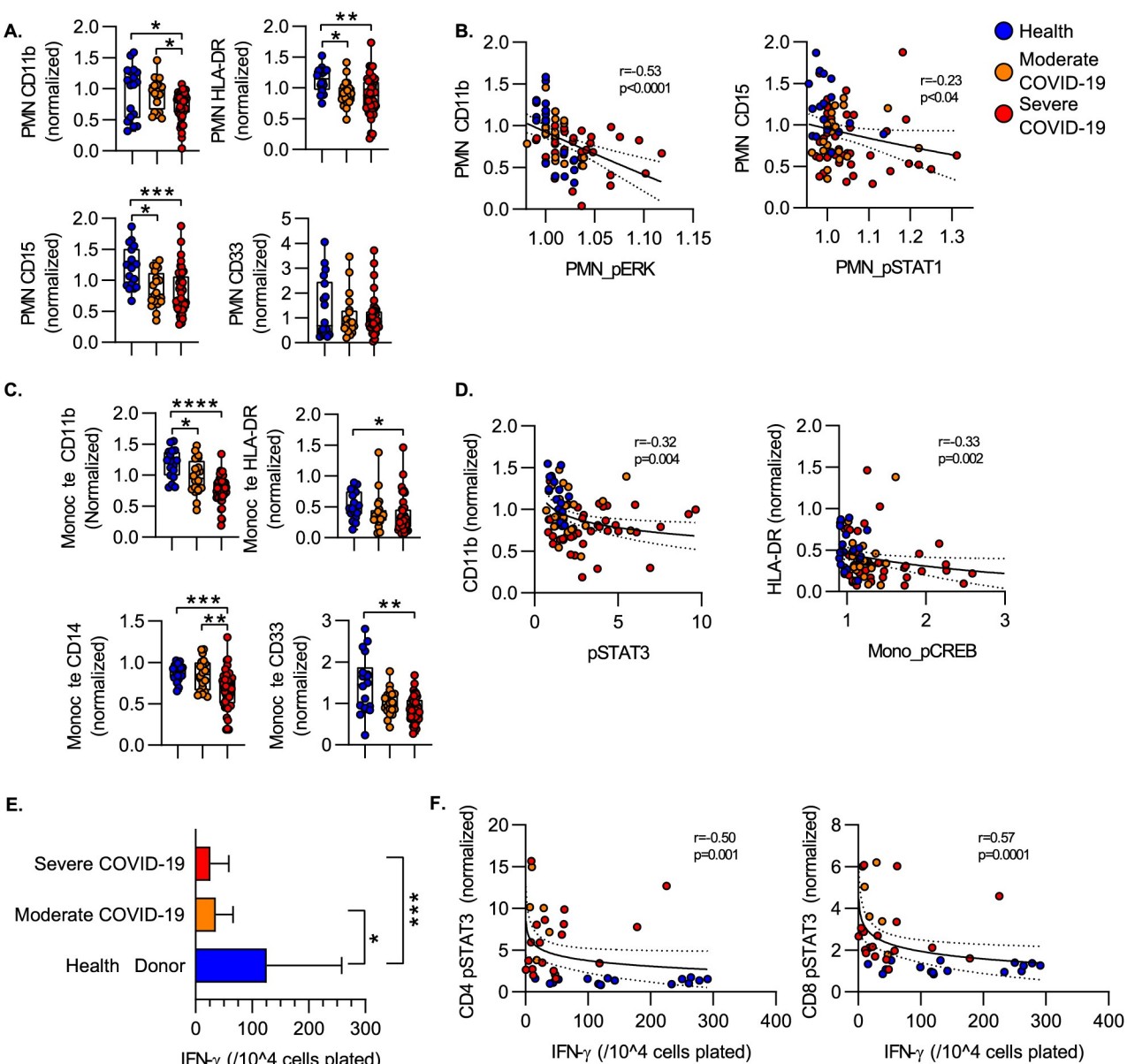

**Fig 6. Activation markers and cytokine production. A)** Levels of surface marker expression on circulating neutrophils. **B)** Correlation between neutrophil surface marker expression level and intracellular phosphoprotein levels. **C)** Levels of surface marker expression on circulating neutrophils. **D)** Correlation between neutrophil surface marker expression level and intracellular phosphoprotein levels. **E)** IFN-γ production measured by ELISpot. PBMC were stimulated with CD3/CD28 antibodies and the number of IFN-γ producing cells enumerated. Bars show median spot number/ $10^4$ cells +/- interquartile range. **F)** Correlation between ELIspot IFN-γ production and intracellular phosphoprotein expression. Correlation plots display results from Spearman correlation test and semilog regression curve (solid line) with 95% confidence intervals (dotted lines).). Box plots represent 25-75th percentile with line defining group median, with all data-points overlying. Error bars extend from minimum to maximum values.* p<0.05, **p<0.01, ***p<0.001, ****p<0.0001 by Dunn's multiple comparisons test after nonparametric (Kruskal-Wallis) ANOVA with correction for FDR<1% (Benjamini, Krieger and Yekutieli).

associated with a wholesale increase in signaling protein phosphorylation in both myeloid and lymphoid cell populations across multiple signaling pathways. In all cell populations there was a strong, <u>positive</u> correlation between an activated phosphoproteome signature and circulating inflammatory cytokine levels. In particular, STAT3 activation was tightly correlated with

plasma IL-6 levels. In myeloid cells signaling protein phosphorylation levels were <u>negatively</u> correlated with surface markers of activation and maturation. Similarly, in T cells, <u>increased</u> STAT3 phosphorylation was specifically associated with <u>defects</u> in IFN-γ production. These data reconcile the coincident inflammatory cytokinemia and functional immunosuppression induced by COVID-19 and suggest modulation of STAT3 as a potential therapeutic avenue to restore T cell function during severe COVID-19.

Consistent with past studies [16–18], we found that COVID-19 was associated with an increase in neutrophil frequency and a concomitant decrease in other circulating leukocyte populations including monocytes, basophils and both CD4 and CD8 T cells. The increased neutrophil frequency was associated with a decreased expression of surface markers of maturation including CD11b, HLA-DR and CD15. COVID-19 is associated with accumulation of peripherally circulating immature low-density neutrophils (LDN) [14], previously characterized to have low-to-intermediate surface expression of CD11b [13] and HLA-DR [13–15]. Similarly, CD15 is a marker of neutrophil antimicrobial function, with decreased CD15 expression being associated with high TB microbial burden and treatment failure [31]. We also found decreased expression of monocyte maturation markers including CD33, CD14 and HLA-DR. This observation is consistent with studies showing that IL-6 is a suppressive cytokine to monocyte CD33 and CD14 levels [32]. This shift in surface marker phenotype likely reflects emergency hematopoiesis [33] with mobilization of immature neutrophils from the bone-marrow driven both by margination of mature neutrophils to the lungs and mobilization of immature cells by inflammatory cytokines including IL-6 and IL-8 [34].

To our knowledge this is the first description of the signaling phosphoproteome in peripheral blood leukocytes during acute infection with SARS-COV2. COVID-19 was associated with an overall increase in signaling protein phosphorylation, and in no case was there a decreased protein phosphorylation in COVID-19 vs. controls. We found that circulating leukocyte phosphoprotein signature alone can distinguish subjects with COVID-19 from healthy donor controls. In all cell populations, STAT3 was the predominant signaling pathway activated during COVID-19. STAT3 is the predominant downstream target of the IL-6 receptor [28, 30]. COVID-19 induces high levels of circulating IL-6, and we found that phospho-STAT3 levels were tightly correlated with plasma IL-6 levels. Although we detected upregulation of multiple signaling pathways, the specific increase in STAT3 phosphorylation during COVID-19 was striking. Recent data preprint data from Feyaerts et al. provide an additional perspective [35]. These authors undertook a multiomic study of subjects with COVID-19 and healthy controls. These authors stimulated cells ex-vivo and measured the signaling response, finding that COVID dampened activation of the NF-κB and JAK/STAT pathways. In contrast to our data and data from Bronte et al. [36], these authors find little upregulation of STAT3 in patients with COVID. In part these discrepancies may reflect different patient population. The subjects of our study were all recruited within 72 hours of hospital presentation and includes preponderance of severely ill patients. Feyaerts at al may have recruited a cohort of patients throughout a more prolonged time course, and their data may reflect a change in the COVID-19 signaling phosphoproteome as the disease progresses.

STAT3 is an evolutionarily primordial signaling molecule with pleiotropic effects in both immune and non-immune tissues including mediating lymphocyte differentiation and function, myeloid cell activation, emergency hematopoiesis and the hepatic acute phase response [37, 38]. We found near universal increases in STAT3 phosphorylation during both moderate and severe COVID-19, likely reflecting persistent STAT3 activity in both monocytes and lymphocytes. Although STAT3 is canonically associated with immune cell activation and transcription of pro-inflammatory genes, exaggerated STAT3 activation can results in immunosuppression [38–40]. Genetic STAT3 gain of function (GOF) mutations result in

persistent, chronic STAT3 activity, and phenocopies the changes in peripheral leukocyte populations seen in severe COVID-19, with decreased circulating levels of pDC, NK cells and Th17 lymphocytes [41, 42]. These patients are also at higher risk of disseminated mycobacterial infection [41, 43], concerning for defects in antigen presenting cell function. In part the immunosuppressive effects of STAT3 reflects fine-tuning of the STAT3 transcriptional response by SOCS3, which suppresses STAT3 immunosuppressive activity [44]. In the absence of SOCS3 activity, STAT3 activation induced by IL-6 can lead to decreased expression of MHC-II molecules and co-stimulatory molecules on the surface of antigen presenting cells. In-vitro data support this observation, with STAT3 repressing transcription of pro-inflammatory genes after persistent LPS stimulation. Transcriptome studies have reported decreased levels of SOCS-3 in myeloid cells during COVID-19 [45], which could lead to an immunosuppressive response to IL-6-mediated STAT3 activation during COVID-19. This hypothesis is consistent with studies demonstrating defects in dendritic cell response during COVID-19 [23, 45], and other data demonstrating a strong negative correlation between plasma IL-6 levels and monocytes HLA-DR expression [46]. Our data support this hypothesis, including the depressed monocyte surface expression of HLA-DR and CD86 reported here, and our previously published data demonstrating that COVID-19 is associated with decreased LPS-induced TNF-alpha production by peripheral blood mononuclear cells.

STAT3 has also been also been shown to inhibit the Type-I IFN antiviral response. COVID-19 is associated with increased circulating levels of IFN-alpha [26], and our analysis did measure a 2-fold increase in IFN-alpha in severe COVID vs. healthy donors (although these data did not meet the threshold for a 1% FDR, see S4 Table "Immunologic Features Defined by CyTOF"). Transcriptomic studies detected expression of Interferon-stimulated genes (ISG) in mild and moderate COVID-19, but an impaired Type-I interferon response in severe COVID-19 [47, 48]. Others reproduced these results, finding that severe influenza was associated with upregulation of ISG which was not seen in severely ill COVID-19 patient [26]. STAT3-mediated inhibition of the IFN response is independent of STAT3 transcriptional activity and is mediated through negative regulation of the cytosolic dsRNA receptor MDA5 [47]. MDA5 serves as a pattern recognition receptor for SARS-CoV2 underlying the IFN response during COVID-19 [49], and the decreased ISG expression during COVID-19 may result from increased STAT3 mediated inhibition of MDA5 activity.

We acknowledge several limitations in our study. Foremost, our study design provides limited clinical information for our cohort of COVID-19 subjects, and we have not associated our findings with clinical outcomes from COVID-19. We have segregated our patients into "moderate" and "severe" COVID-19 disease, and we do report clinical outcomes including ICU admission, ventilator days and hospital length of stay are segregate based on disease severity, and we report differences in signaling phosphoproteome between our cohorts, but our limited sample size and limited clinical data constrains our ability to do a controlled analysis of the relationship between specific phosphoproteome features and clinical outcome. We predict that phospho-STAT3 levels on hospital admission may have prognostic significance, but this hypothesis will require further investigation. In addition, we recognize that there is a trend toward increased age in the severe vs. moderate cohort that may confound our analysis. Although the difference in ages did not reach statistical significance, the uncontrolled variance in age distribution remains a limitation of our analysis and it is possible that some of the effects of we ascribe to severe COVID reflect age-related changes independent of COVID-19. We also have utilized ex-vivo CD3 and CD28 stimulated IFN-γ production as measured by the ELISpot assay as a proxy for immunocompetence. We have previously demonstrated in patients with sepsis and patients with COVID-19 that defects in IFN-γ production are associated with worse outcomes. Our data analysis approach also has limitations. We have analyzed our CyTOF data

by manual gating of canonical markers. We have only compared signaling protein phosphory-lation across a limited set of major circulating populations. We recognize that by aggregating together leukocyte subpopulations, we may obscure differences in phosphoprotome activation in leukocyte subpopulations. We pursued this like of investigation, unfortunately, our ability to analyze subpopulations of cells was largely limited by a limited event frequency. This largely reflects the practical reality of a whole blood fixation protocol. Because we have not purified PBMC, we have a relatively decreased number of lymphocytes in the preparation. This, com-bined with the relative cytopenia caused by COVID-19 has left us with a relatively small num-ber of cells for downstream analysis. We attempted both manual gating approaches and automated clustering algorithms (including t-distributed stochastic neighbor embedding and Uniform Manifold Approximation and Projection for Dimension Reduction) and found that we had insufficient numbers of events to reliably identify consistent populations that could be analyzed for signaling protein levels. The data utilized for this study will be made available in publicly accessible database, and we encourage collaborating investigators to undertake further analysis to elucidate these differences.

Given these limitations, these data demonstrate that COVID-19 causes an acute dysregula-tion of the signaling landscape of circulating leukocytes, defined primarily by exaggerated phosphorylation of STAT3. In both myeloid and lymphoid cell populations, pSTAT3 levels are correlated with evidence of functional immunosuppression, suggesting STAT3 blockade as a potential therapeutic pathway to restore immunocompetence during severe COVID-19. Con-sistent with this, recent studies have demonstrated efficacy for the JAK/JAK2 inhibitor Bariciti-nib in the treatment of COVID-19, a result that would have been anticipated by our data [36, 50]. We further find evidence that STAT3 activation is associated with defects in IFN-γ pro-duction by T cells. It will be important to pursue mechanistic studies using STAT3 inhibitors such as Nababuscin or inhibitors of the signaling pathway upstream of STAT3 such as Baraciti-nib to demonstrate a causal relationship beween STAT3 activity and decreased IFN-γ produc-tion. It would also be extremely useful to measure the effects of treatment with Baricitinib on both STAT3 levels and ex-vivo cytokine production by ELISpot. Prior reports have advocated STAT3 inhibition to restrain inflammation-mediated tissue damage [10, 12]. These reports extend from a hypothesized role for STAT3 in the immunopathology of COVID-19 through the IL-6 amplifier pathway, whereby STAT3 and NFκB synergize to drive inflammatory gene transcription locally in non-immune tissue, driving inflammation-mediated tissue injury [51]. We propose that dysregulated STAT3 activation reconciles local inflammation-mediated tissue damage and systemic functional immunosuppression, and suggests that pharmacologic inhibi-tion of STAT3 may simultaneously restrain local inflammation-mediated tissue damage while restoring systemic antiviral immune function.

## Supporting information

**S1 Table. Antibody panels for surface and intracellular staining.**
(PDF)

**S2 Table. Cohort demographics and clinical information.**
(PDF)

**S3 Table. Cohort clinical laboratory values.**
(PDF)

**S4 Table. Immunologic features defined by CyTOF.**
(PDF)

**S5 Table. Soluble marker levels.**
(PDF)

**S1 Fig. CyTOF gating schema.**
(PDF)

**S2 Fig. Hierarchical clustering of soluble markers.**
(PDF)

## Acknowledgments

We would like to acknowledge our colleagues working at Barnes Jewish Hospital and the Missouri Baptist Medical Center for the excellent care they provided to the patients in this study.

## Author Contributions

**Conceptualization:** Isaiah R. Turnbull, Anja Fuchs, Kenneth E. Remy, Michael P. Kelly, Monty B. Mazer, Jennifer M. Leonard, Mark H. Hoofnagle, Marco Colonna, Richard S. Hotchkiss.

**Data curation:** Anja Fuchs.

**Formal analysis:** Isaiah R. Turnbull, Anja Fuchs.

**Funding acquisition:** Isaiah R. Turnbull.

**Investigation:** Isaiah R. Turnbull, Anja Fuchs, Kenneth E. Remy, Michael P. Kelly, Elfaridah P. Frazier, Sarbani Ghosh, Shin-Wen Chang, Monty B. Mazer, Annie Hess, Jennifer M. Leonard, Mark H. Hoofnagle, Richard S. Hotchkiss.

**Supervision:** Isaiah R. Turnbull.

**Writing – original draft:** Isaiah R. Turnbull.

**Writing – review & editing:** Kenneth E. Remy, Michael P. Kelly, Elfaridah P. Frazier, Monty B. Mazer, Annie Hess, Jennifer M. Leonard, Mark H. Hoofnagle, Marco Colonna, Richard S. Hotchkiss.

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
