## [Decision Letter · Decision Letter 0]

31 Dec 2021

PONE-D-21-38962Dysregulation of the Leukocyte Signaling Landscape during Acute COVID-19PLOS ONE

Dear Dr. Turnbull,

Thank you for submitting your manuscript to PLOS ONE. After careful consideration, we feel that it has merit but does not fully meet PLOS ONE’s publication criteria as it currently stands. Therefore, we invite you to submit a revised version of the manuscript that addresses the points raised during the review process. In addition to the reviewer’s comment appended below, I have concerns about lack of inclusion of appropriate controls during experimentations. Please pay particular attention to this aspect while addressing the comments raised by the reviewers.

We look forward to receiving your revised manuscript.

Kind regards,

Sumit Kumar Hira, Ph.D.

Academic Editor

PLOS ONE

Journal Requirements:

"IRT and RSH have pending intellectual property related to the ELISpot assay."

Reviewers' comments:

Reviewer's Responses to Questions

**Comments to the Author**

1. Is the manuscript technically sound, and do the data support the conclusions?

Reviewer #1: Yes

Reviewer #2: Partly

2. Has the statistical analysis been performed appropriately and rigorously? 

Reviewer #1: Yes

Reviewer #2: Yes

3. Have the authors made all data underlying the findings in their manuscript fully available?

Reviewer #1: Yes

Reviewer #2: Yes

4. Is the manuscript presented in an intelligible fashion and written in standard English?

Reviewer #1: Yes

Reviewer #2: Yes

5. Review Comments to the Author

Reviewer #1: In their study, the authors investigated the signaling pathways related to dysregulated immunities in COVID-19 using the methods of mass cytometry, multiplex cytokine detection and IFN-gamma ELISPOT assay. They found that the levels of phosphorylated STAT1 and STAT3 in multiple lymphocyte populations correlated significantly with plasma levels of several cytokines. The JAK/STAT signaling pathway has been suggested to be a key player in virus caused cytokine storm. IL-6 is known to be a major activator of JAK/STAT pathway. This study confirmed that the plasma IL-6 levels correlated with the activation of STAT pathway in COVID-19 patients early after admission.

Specific comments and questions are listed as follows.

1. 81 subjects were enrolled in this study, including 43 severe cases, 20 moderate cases and 18 healthy individuals. However, it seems the number of patients included in each major assay is different (as shown in Figure 1A). The authors need to explain why.

2. The severe cases tends to be older than the moderate cases and the healthy individuals. Could this be a confounding factor that may affect the comparisons among different groups?

3. The authors mentioned that a common reference was employed to normalize the phosphoproteome data. I suggest the authors to specify the reference.

4. The release of IFN-gamma was induced by anti-CD3 and anti-CD28 mAbs. However, its relevance with antiviral immunities is not clear.

5. Elevated levels of phosphorylated CREB and PLCgamma2 were observed in T cells of severe COVID-19 cases (Fig.4). As these two factors may participate TCR signaling, correlation analyses between levels of pCREB/pPLCgamma2 and IFN-gamma releasing can help to understand the COVID-19 related compromise of T cell response.

6. The description about Wuhan in the introduction section is not accurate. Wuhan city is the capital of Hubei province (central China).

Reviewer #2: Turnbull et al. analyze a cohort of patients (n=63) with moderate and severe COVID-19 as well as 18 healthy controls using multiplexed serum proteomics and mass cytometry focusing on the phosphoproteome in the main leukocyte populations. Furthermore, they use an Enzyme-Linked Immunospot assay to investigate the T cell activation potential in a non-antigen specific way.

The authors describe a profound cytopenia of certain leukocyte subsets, which is consistent with previously published studies. They then analyze a broad variety of phosphoproteins in the main leukocyte subsets. They claim to identify increased pSTAT3 and pSTAT1 signaling in COVID-19 patients in different leukocyte subsets compared to healthy controls. Furthermore, they show that the pSTAT3 and pSTAT1 correlate with circulating IL-6 and CXCL10 as well as reduced numbers of IFN-gamma producing T cells upon CD3/CD28 stimulation.

The aim of analyzing the phosphoproteome of circulating leukocytes in COVID-19 is very relevant to further understand the pathophysiology of severe COVID-19. The approach the authors took and the cohort design seems very suited to this. To further strengthen some of the authors conclusions, I would have the following suggestions:

- Regarding CyTOF analysis of the phosphorylated signaling proteins, it seems surprising that the healthy controls seem to separate in 2 subgroups (it might be good to show the dendrogram for the patient clustering as well) with one having quite high signals in pNFkB and pSTAT5 signals among others. Could this be an indication that the phospho-signal differences are very low? It might be good to show more primary data especially on the STAT3 and STAT1 signals the authors subsequently focus on, to better appreciate how strong the differences are. Furthermore, it might be worth considering having additional controls such as conditions with eg. STAT inhibitors and IL-6 stimulations to make the conclusion stronger.

- Similarly, to further support the hypothesis of IL-6 as the responsible cytokine for the pSTAT3 signals it might be worth looking at different T cell populations which are included in the current panel (memory populations, regulatory T cells) to see if the pSTAT3 signals differ as expected and furthermore if the signal correlates with IL-6R levels (Ridgley et al. Front Imm 2019).

- I agree that it’s very interesting to have functional data complementing the T cell measurements. However, I think it might be difficult to conclude that the reduced number of IFN-gamma producing T cells is an actual defect in T cell function as this might just be an indication of the lymphopenia in these patients or are the numbers corrected for this issue?

Minor points

- For the cohort description, if feasible it might be good to know the time after symptom onset in the different patient groups as this can strongly influence the results (Chevrier et al. Cell Reports Medicine 2021), however it might of course be difficult to extract this information which would be very understandable. The treatment at the time of sampling could also be an important consideration eg. hydrocortisone or tocilizumab could be very relevant to know, if a group of patients would have been treated with the latter, it could also be very interesting to see whether the pSTAT3 levels were to differ in this group.

- An interesting paper for the discussion on the role of IL-6 might be Giamarellos-Bourboulis et al. Cell Host & Microbe 2020.

- Feyaerts et al. bioRxiv 2021 also assessed leukocyte phosphoproteome in COVID-19, however they find rather reduced signaling pathways, maybe it would be good to compare the authors results to these.

- Methods

o I missed the information on how the cells/ml in Figure 2 were calculated?

- Typos

o The numbering for supplementary Figures and Tables was incorrect on a few instances.

o In Figure 3, it should probably read NFkB instead of NKFB

o “In contrast, monocyte phospho-ERK levels were most correlated with plasma levels of IL-8 (Fig. 5)” the exact subfigure is missing

6. PLOS authors have the option to publish the peer review history of their article (what does this mean?). If published, this will include your full peer review and any attached files.

Reviewer #1: No

Reviewer #2: No

---

## [Author Response · Author response to Decision Letter 0]

8 Feb 2022

Reviewer #1: In their study, the authors investigated the signaling pathways related to dysregulated immunities in COVID-19 using the methods of mass cytometry, multiplex cytokine detection and IFN-gamma ELISPOT assay. They found that the levels of phosphorylated STAT1 and STAT3 in multiple lymphocyte populations correlated significantly with plasma levels of several cytokines. The JAK/STAT signaling pathway has been suggested to be a key player in virus caused cytokine storm. IL-6 is known to be a major activator of JAK/STAT pathway. This study confirmed that the plasma IL-6 levels correlated with the activation of STAT pathway in COVID-19 patients early after admission.

Specific comments and questions are listed as follows.

1. 81 subjects were enrolled in this study, including 43 severe cases, 20 moderate cases and 18 healthy individuals. However, it seems the number of patients included in each major assay is different (as shown in Figure 1A). The authors need to explain why.

Thanks for the identifying this oversight. All samples underwent CyTOF analysis, but only a subset underwent either ELISpot or plasma cytokine analysis, based solely on sample volume and lab staffing (there was no selection process). Both the ELISpot assay and the plasma cytokine analysis require more complex initial sample handling (either to run the ELISpot assay or to separate plasma from cellular elements). The CyTOF assay only requires that 1 mL of whole blood be fixed in proteomic stabilizer. We measured function by ELISpot and collected plasma whenever there was adequate sample volume and when lab staffing logistics allowed. We have added text to the methods clarifying the reasons for the variation in subjects across the assay. 

2. The severe cases tends to be older than the moderate cases and the healthy individuals. Could this be a confounding factor that may affect the comparisons among different groups?

We agree that the increased age is a confounder for comparisons across the groups, and that this is a limitation of our study. Although there is a trend toward an increased age in the severe vs. moderately COVID patients, there is no statistically significant difference (moderate: 55.6+/-14.6 vs. severe 60.1+/-18.4, t-test p-value 0.32. Healthy: 52.5+/-18 vs. moderate and severe, ANOVA p-value 0.21). We recognize that this trend still could confound our analysis. Unfortunately we are underpowered for multivariate regression and cannot statistically control for the effects of age in the setting of COVID. We have amended the manuscript to specifically acknowledge the trend toward increased age in the severe vs. moderate cohort and address this as a limitation. 

3. The authors mentioned that a common reference was employed to normalize the phosphoproteome data. I suggest the authors to specify the reference.

The common reference is a common healthy donor blood sample was included in every batch as an internal control across CyTOF run. Blood was isolated from a healthy individual at a single time point, aliquoted and frozen. This sample was included in every as a reference standard. We have amended the manuscript to clarify this process. 

4. The release of IFN-gamma was induced by anti-CD3 and anti-CD28 mAbs. However, its relevance with antiviral immunities is not clear.

We agree that we cannot make draw any conclusion regarding the anti-viral utility of ex-vivo CD3/CD28 stimulated IFN-gamma production. We have previously reported that persistent suppression of ex-vivo CD3/CD28 stimulated IFN-gamma production as measured by ELISpot was associated with increased mortality in patients with COVID-19 (Remy KE et al, Severe immunosuppression and not a cytokine storm characterizes COVID-19 infections. JCI Insight. 2020 Sep 3;5(17):e140329. PMCID: PMC7526441). We acknowledge though these data alone do not ascribe a specific anti-viral effect of IFN-gamma (for example, the defects in T cell IFN-gamma production could leave patients susceptible to other infections). We have revised the discussion to address this topic. 

5. Elevated levels of phosphorylated CREB and PLCgamma2 were observed in T cells of severe COVID-19 cases (Fig.4). As these two factors may participate TCR signaling, correlation analyses between levels of pCREB/pPLCgamma2 and IFN-gamma releasing can help to understand the COVID-19 related compromise of T cell response.

We appreciate the suggestion. We have evaluated the correlation between these features. Unfortunately, with the exception of CD4 pCREB, these correlations did not reach statistical significance and were in general lower than those measured for pSTAT3. (See table below). We have added text to the results section describing these findings.

 CD8_pCREB CD8_PLCG2 CD4_pCREB CD4_PLCG2

 IFN (/10^4 cells plated) Correlation Coefficient -.366* -.251 -.460** -.241

 Sig. (2-tailed) .020 .119 .003 .134

 N 40 40 40 40

6. The description about Wuhan in the introduction section is not accurate. Wuhan city is the capital of Hubei province (central China).

Thank you, we have corrected the error.

Reviewer #2: Turnbull et al. analyze a cohort of patients (n=63) with moderate and severe COVID-19 as well as 18 healthy controls using multiplexed serum proteomics and mass cytometry focusing on the phosphoproteome in the main leukocyte populations. Furthermore, they use an Enzyme-Linked Immunospot assay to investigate the T cell activation potential in a non-antigen specific way.

The authors describe a profound cytopenia of certain leukocyte subsets, which is consistent with previously published studies. They then analyze a broad variety of phosphoproteins in the main leukocyte subsets. They claim to identify increased pSTAT3 and pSTAT1 signaling in COVID-19 patients in different leukocyte subsets compared to healthy controls. Furthermore, they show that the pSTAT3 and pSTAT1 correlate with circulating IL-6 and CXCL10 as well as reduced numbers of IFN-gamma producing T cells upon CD3/CD28 stimulation.

The aim of analyzing the phosphoproteome of circulating leukocytes in COVID-19 is very relevant to further understand the pathophysiology of severe COVID-19. The approach the authors took and the cohort design seems very suited to this. To further strengthen some of the authors conclusions, I would have the following suggestions:

- Regarding CyTOF analysis of the phosphorylated signaling proteins, it seems surprising that the healthy controls seem to separate in 2 subgroups (it might be good to show the dendrogram for the patient clustering as well) with one having quite high signals in pNFkB and pSTAT5 signals among others. Could this be an indication that the phospho-signal differences are very low? It might be good to show more primary data especially on the STAT3 and STAT1 signals the authors subsequently focus on, to better appreciate how strong the differences are. Furthermore, it might be worth considering having additional controls such as conditions with eg. STAT inhibitors and IL-6 stimulations to make the conclusion stronger.

We appreciate the reviewer calling our attention to this observation regarding the clustered heatmap data (Figure 3). We agree that in this heatmap it appears that for some features there are 2 population of healthy controls. The differences are amplified in the heat map as in this figure the color ramping for each feature is normalized by row. To control for this, we have included the primary data for each individual in Figure 4 to explicitly show the values, including STAT1 and STAT3. These are the same data that are used to make the heatmap, but in Figure 4 we explicitly show the individual values for each feature. We did not include these data for NFkB as there were no statistically significant difference in NFkB across the groups (likely due to a small effect size as the reviewer correctly elucidated). Similarly there was no difference in STAT5 across these groups (see figure below by example). We have edited the manuscript to address this issue.

We are pursuing additional studies using ex-vivo stimulation such as IL-6 and STAT inhibitors (in both patients with COVID, and also in other populations of critically ill patients as our critically ill COVID patient volume is fortunately decreasing. Although these important mechanistic studies are underway, this has requires a completely new clinical protocol for sample acquisition and processing. For the current study we did not have adequate sample volume for additional ex-vivo studies and the sample handling protocol also did not allow for these additional studies. We have acknowledged this as a limitation and included this as an important future direction in the discussion. 

- Similarly, to further support the hypothesis of IL-6 as the responsible cytokine for the pSTAT3 signals it might be worth looking at different T cell populations which are included in the current panel (memory populations, regulatory T cells) to see if the pSTAT3 signals differ as expected and furthermore if the signal correlates with IL-6R levels (Ridgley et al. Front Imm 2019).

We strongly agree with the reviewer and this was our initial intention when analyzing these data. Unfortunately several factors have worked against us achieving this goal. First, we use a whole blood fixation protocol. Because we have not purified PBMC, we have a relatively decreased number of lymphocytes in the preparation. This, combined with the relative cytopenia caused by COVID-19 has left us with a relatively small number of cells for downstream analysis. We attempted both manual gating approaches and automated clustering algorithms (tSNE, UMAP) and found that we had insufficient numbers of events to reliably identify consistent populations that could be analyzed for signaling protein levels. This was further complicated by the batch effects associated with CyTOF. After much effort, we found that we could not generate reliable, reproducible data regarding the subpopulations of T cell using our approach. We further address this as a limitation in the discussion. 

- I agree that it’s very interesting to have functional data complementing the T cell measurements. However, I think it might be difficult to conclude that the reduced number of IFN-gamma producing T cells is an actual defect in T cell function as this might just be an indication of the lymphopenia in these patients or are the numbers corrected for this issue?

To control for this issue the ELISpot data are corrected for the number of lymphocytes plated and expressed as spot forming units (SFU)/10^4 cells plated.

Minor points

- For the cohort description, if feasible it might be good to know the time after symptom onset in the different patient groups as this can strongly influence the results (Chevrier et al. Cell Reports Medicine 2021), however it might of course be difficult to extract this information which would be very understandable. The treatment at the time of sampling could also be an important consideration eg. hydrocortisone or tocilizumab could be very relevant to know, if a group of patients would have been treated with the latter, it could also be very interesting to see whether the pSTAT3 levels were to differ in this group.

Unfortunately the clinical data regarding pharmaceutical treatments are not available for these samples. Also of note, these samples were collected prior to the routine use of Tocilzumab at our institution, (just noting the rapid change in treatment protocols for COVID!). We have added material to the discussion addressing the role of Tocilizumab and the potential utility of this approach in both measuring an effect of this agent or other aging. 

- An interesting paper for the discussion on the role of IL-6 might be Giamarellos-Bourboulis et al. Cell Host & Microbe 2020.

- Feyaerts et al. bioRxiv 2021 also assessed leukocyte phosphoproteome in COVID-19, however they find rather reduced signaling pathways, maybe it would be good to compare the authors results to these.

We have added these important papers to the discussion.

- Methods

o I missed the information on how the cells/ml in Figure 2 were calculated?

The cell numbers were enumerated using a hemocytometer as part of the CyTOF sample processing. This has been clarified in the methods under sample processing.

- Typos

o The numbering for supplementary Figures and Tables was incorrect on a few instances. 

Thank you, we will carefully review.

o In Figure 3, it should probably read NFkB instead of NKFB

Thank you, corrected. 

o “In contrast, monocyte phospho-ERK levels were most correlated with plasma levels of IL-8 (Fig. 5)” the exact subfigure is missing

Thank you, corrected.

---

## [Decision Letter · Decision Letter 1]

22 Feb 2022

Dysregulation of the Leukocyte Signaling Landscape during Acute COVID-19

PONE-D-21-38962R1

Dear Dr. Turnbull,

We’re pleased to inform you that your manuscript has been judged scientifically suitable for publication and will be formally accepted for publication once it meets all outstanding technical requirements.

Kind regards,

Sumit Kumar Hira, Ph.D.

Academic Editor

PLOS ONE

Additional Editor Comments (optional):

Reviewers' comments:

Reviewer's Responses to Questions

**Comments to the Author**

1. If the authors have adequately addressed your comments raised in a previous round of review and you feel that this manuscript is now acceptable for publication, you may indicate that here to bypass the “Comments to the Author” section, enter your conflict of interest statement in the “Confidential to Editor” section, and submit your "Accept" recommendation.

Reviewer #1: All comments have been addressed

Reviewer #2: All comments have been addressed

2. Is the manuscript technically sound, and do the data support the conclusions?

Reviewer #1: Yes

Reviewer #2: Yes

3. Has the statistical analysis been performed appropriately and rigorously? 

Reviewer #1: Yes

Reviewer #2: Yes

4. Have the authors made all data underlying the findings in their manuscript fully available?

Reviewer #1: Yes

Reviewer #2: Yes

5. Is the manuscript presented in an intelligible fashion and written in standard English?

Reviewer #1: Yes

Reviewer #2: Yes

6. Review Comments to the Author

Reviewer #1: Previous comments and questions have fully addressed. The manuscript has also been revised accordingly. I have no further comment.

Reviewer #2: The authors answered my comments satisfactorily. I would still think that the additional controls would be helpful, however I also understand the difficulties and by mentioning the caveats in the manuscript the authors have addressed the issue.

Minor points

- The authors mentioned that they tried unsupervised analysis methods and mention tSNE and UMAP as clustering approaches. They are however dimensionality reduction methods for visualization and not clustering approaches per se, eg. of the latter would be FlowSOM and Phenograph.

7. PLOS authors have the option to publish the peer review history of their article (what does this mean?). If published, this will include your full peer review and any attached files.

Reviewer #1: No

Reviewer #2: No

---

## [Editor Report · Acceptance letter]

5 Apr 2022

PONE-D-21-38962R1 

Dysregulation of the Leukocyte Signaling Landscape during Acute COVID-19 

Dear Dr. Turnbull:

I'm pleased to inform you that your manuscript has been deemed suitable for publication in PLOS ONE. Congratulations! Your manuscript is now with our production department. 

Kind regards, 

on behalf of

Dr. Sumit Kumar Hira 

Academic Editor

PLOS ONE